# Group Robustness via Discounted Rank Upweighting

## Abstract

Recent work has shown that standard training via empirical risk minimization (ERM) can produce models that achieve high accuracy on average but low accuracy on underrepresented groups due to the prevalence of spurious features. A predominant approach to tackle this group robustness problem minimizes the worst group error (akin to a *minimax* strategy) on the training data with the expectation that it will generalize well to unseen test data. However, this is often suboptimal, especially when the out-of-distribution (OOD) test data contains previously unseen groups. Inspired by ideas from the information retrieval and learning-to-rank literature, this paper first proposes to use Discounted Cumulative Gain (DCG) as a metric of model quality for facilitating better hyperparameter tuning and model selection. Being a ranking-based metric, DCG weights multiple poorly-performing groups (instead of considering just the group with the worst performance). As a natural next step, we build on our results to propose a ranking-based training method called **Discounted Rank Upweighting (DRU)** which differentially reweights a ranked list of poorly-performing groups in the training data to learn models that exhibit strong OOD performance on the test data. Results on several synthetic and real-world datasets highlight the superior generalization ability of our group-ranking-based (akin to *soft-minimax*) approach in selecting and learning models that are robust to group distributional shifts.

## 1 Introduction

Text data are naturally split into groups in many machine learning contexts, e.g., sentiment classification with reviews from different users or a personalized dialogue system. In both these examples, a group corresponds to a user. In other contexts, such as online toxicity detection, the groups might be implicit, e.g., user demographics, and may require annotation. More broadly, consider the scenario where training examples are stratified non-uniformly into groups. Our goal is to build a model for this scenario that generalizes to all groups by providing comparable classification accuracies—a key objective of deploying robust and fair machine learning models (Dwork et al., 2012; Hardt et al., 2016; Kleinberg et al., 2016).

Recent work on robust and equitable machine learning has shown that the traditional approach of minimizing the average training error, also known as empirical risk minimization (ERM), can be suboptimal for this grouped data setting. ERM produces models that achieve low test error on average but incur high errors on underrepresented groups in the data, which raises serious ethical and fairness concerns. One of the main reasons ERM conceals poor performance on minority groups behind a vastly superior average accuracy is its reliance on spurious relationships between labels and some features in the majority groups to achieve high average accuracy (Hovy & Søgaard, 2015; Blodgett et al., 2016; Tatman, 2017; Hashimoto et al., 2018; Duchi et al., 2021). Such correlations between labels and features are nonexistent or present with an opposite sign in the minority (or new) groups. This leads to ERM severely underperforming on these groups while overfitting to the majority groups (Buolamwini & Gebru, 2018; Koh et al., 2021).

Prior research tackles spurious correlation by building models with low worst-group error on the training dataset. One such prominent model, Group Distributional Robust Optimization (Group DRO), seeks to minimize the worst group's training loss (Sagawa et al., 2019). While Group DRO has shown promising performance compared to ERM on some benchmark datasets, it is known to perform poorly when the different groups contain varying amounts of predictive signal (Koh et al., 2021). Group DRO assumes the

test groups are all seen during training and each group has a distribution that is invariant between training and test. Following this assumption, the worst group in test is also likely the worst in training. Thus, Group DRO extends the minimax distributional robust optimization (DRO) framework (Namkoong & Duchi, 2016) to groups. However, this assumption does not always hold in reality, and it is especially problematic in *domain-generalization* scenarios where the test data contains previously unseen out-of-distribution (OOD) groups that do not overlap with the training or validation data.[1] Moreover, unlike datasets with few groups and clear identification of spurious features by construction (analogous to a controlled experiment), e.g., WaterBirds (Wah et al., 2011), spurious features can be hard to locate in naturally grouped datasets, as they usually are not present exclusively in a subset of groups. For instance, in tasks such as sentiment classification of user reviews, potential spurious features such as the writing style can be present in all the groups to varying degrees. We conjecture that differentially reweighting the various groups will help mitigate the impact of spurious features and help us identify robust predictive patterns in the data.

In this paper we draw on ideas from the learning-to-rank literature to provide a more effective solution to the group distributional robustness problem. Specifically, we *rank* and *reweight* different groups based on their training errors (as opposed to considering just the worst-performing group as done by Group DRO or DRO). To summarize, we make two key contributions in this paper: **We develop methods that reweight groups based on the (reverse) ranking of their classification accuracy to 1) choose hyperparameters and perform model selection, and 2) train the model.**

First, we use the Discounted Cumulative Gain (DCG) (Järvelin & Kekäläinen, 2002) metric from the information retrieval and learning-to-rank literature to rank and then reweight several *poorly* performing groups to inform model selection. DCG allows us to consider the validation performance across more than one group while choosing hyperparameters, thus lowering the risk of overfitting. Further, the DCG metric is less prone to having ties in hyperparameter choices, leading to statistically identified models. Next, we turn to the task of developing a novel training method for group distributional robustness. Borrowing intuition from our use of DCG for model selection, we propose a new robust training method called Discounted Rank Upweighting (DRU). DRU iteratively upweights groups during each training epoch based on that group's classification accuracy ranking.

At a high level, our proposed approach can be seen as a *soft-minimax* strategy, which *smooths* the predictive signal from multiple poorly-performing groups by weighting them based on their accuracy-based ranked order. As we show later, both DCG-based model selection and DRU-based model training outperform multiple state-of-the-art methods for group distributional robustness on several synthetic and real-world benchmark datasets.

The rest of the paper is organized as follows. Section 2 discusses the related work. Section 3 describes the preliminaries, including problem setup and baseline methods. Section 4 introduces our approach for achieving group distributional robustness, incorporating DCG-based ranking and reweighting for model selection and presenting a novel training method, DRU. Then, in sections 5 and 6, we present the experiment results for DCG-based model selection and DRU for model training, respectively. We conclude in Section 7.

## 2 Related Work

This paper focuses on group distributional robustness, i.e., training models to generalize well across groups. There are other notions of robustness in machine learning, e.g., adversarial robustness or the study of long-tailed distributions, but they are beyond the scope of this paper.

### 2.1 Group Distributional Shifts

There are a couple of ways to split data into groups based on prior work. First, groups can occur in data organically based on the data collection procedure. For example, all reviews by a given user can be assembled into a group, or all the images taken from a particular camera can constitute a group. All the

---

[1]It turns out that the online algorithm that implements Group DRO in the paper Sagawa et al. (2019) does consider multiple groups, unlike the theory proposed in the paper, but it makes strong parametric assumptions and weights the different groups exponentially, which often leads to suboptimal performance on the test dataset.

items in one group share similar characteristics and are assumed to follow the same data-generating process. These organic groups can be further divided into sub-populations based on meta-information about each group. For instance, the user groups can be divided into sub-groups based on the demographic constitution of those groups. Similarly, images taken from the same camera can be divided into sub-groups based on the photographer's identity. The data can also be split into groups based on the interaction between the output label and a spurious feature, e.g., Waterbirds (Wah et al., 2011), CelebA (Liu et al., 2015), and MultiNLI (Williams et al., 2017) datasets.

Given the importance and prevalence of the grouped data setting, several algorithms have been developed for removing disparity in performance across the different groups. Some popular algorithms include Group Distributionally Robust Optimization (Group DRO), which directly minimizes the worst group's regularized error during model training (Hu et al., 2018; Sagawa et al., 2019). Invariant Risk Minimization (IRM) penalizes the distributions of learned representations with different optimal linear classifiers (Arjovsky et al., 2019). Both Group DRO and IRM require group annotation at training time. Recently, an approach called Just Train Twice (JTT) has been proposed that does not require group information at training time (Liu et al., 2021). JTT instead just upweights misclassified examples and retrains the model. It has been demonstrated to provide superior performance to Group DRO or IRM.

## 2.2 Learning to Rank

It is a subfield of the information retrieval literature which aims to build systems that can accurately retrieve top $k$ documents from a document database. Essentially, it involves ranking the documents in a database based on their content. The common evaluation measures used in this literature include Mean Average Precision (MAP), Discounted Cumulative Gain (DCG), and (Normalized) Discounted Cumulative Gain ((N)DCG), and (N)DCG at $k$ (Järvelin & Kekäläinen, 2000; Järvelin & Kekäläinen, 2002).

**DCG at** $k$ simply adds up the scores earned at each position with inverse logarithm weights up to the $k^{th}$ document, i.e.,

$$\text{DCG@k} = \sum_{i=1}^{k} \frac{\text{Score}_{(i)}}{\log_2(i+1)} \tag{1}$$

While our approach is inspired by learning-to-rank, the major difference is that in information retrieval literature, higher weights are assigned to the **higher**-ranked items (e.g., most relevant documents), while in our setting, higher importance weights are given to **lower**-ranked groups (i.e., worst performing groups). To the best of our knowledge, this is the first work that uses a ranking-based approach to facilitate a *soft-minimax* strategy of training machine learning models with group distributional robustness.

## 3 Preliminaries

We consider the standard supervised learning setup of classifying an input $x \in \mathcal{X}$ as a label $y \in \mathcal{Y}$. We assume that the training data comprises of $m_{train}$ groups from a set $\mathcal{G}$ where each group $g \in \mathcal{G}$ consists of $n_g$ data points from a probability distribution $P_g(\mathcal{X}, \mathcal{Y})$. In addition to the feature $x_j$ and label $y_j$, each training example $j$ is also annotated with the subpopulation/group $g_j \in \mathcal{G}$ that it belongs to. To summarize, the training dataset contains $n_{train}$ samples with group annotations in the format $\{(x_1, y_1, g_1), \ldots, (x_{n_{train}}, y_{n_{train}}, g_{n_{train}})\}$. Our goal is to learn a model $f_\theta : \mathcal{X} \times \mathcal{G} \rightarrow \mathcal{Y}$ parameterized by $\theta \in \Theta$. The group loss for group $g$ is the average loss over all examples in $g$, and we denote it as $l_g(\theta) = \mathbb{E}_{(x,y) \sim P_g(\mathcal{X}, \mathcal{Y})} \mathcal{L}(x, y; f_\theta)$, for a loss function $\mathcal{L}$ and a machine learning model $f_\theta$.

*This paper assumes no group overlap between OOD test and training/validation sets. This is a more challenging setting than the alternative scenario commonly considered in the prior literature, where the test set only contains new proportions of groups but no previously unseen groups.* The performance evaluation metric for a robust model under group distribution shift is the OOD test set accuracy. More concretely, it is preferable to have a model with high worst-group accuracy on the OOD test data, but that does not sacrifice the average accuracy significantly.

### 3.1 Baseline Methods

We compare our approach against several competitive baselines as described below. All the methods (including our approach) use the same base learner $f_\theta$—a finetuned DistilBERT model (Sanh et al., 2019). We describe the hyperparameter choices and other technical details of the various methods later in the paper.

**Empirical Risk Minimization (ERM):**  This is the standard training method that trains models to minimize the average training loss. The method doesn't take any group information into consideration while training the model.

**Group Distributionally Robust Optimization (Group DRO):**  Group DRO uses distributionally robust optimization to explicitly minimize the loss on the worst-case domain (or group) during training. We operationalize Group DRO by using the online algorithm provided by Sagawa et al. (2019).

**Just Train Twice (JTT):**  As briefly described earlier, JTT is a recently proposed approach that requires no group annotations and has shown superior performance over Group DRO and its variations on several challenging benchmark applications (Liu et al., 2021). JTT involves a two-stage training approach which first trains a standard ERM model for several epochs and then trains a second model that upweights the training examples that the first model has misclassified.

## 4 Methods

As just described, the goal of group distributional robustness is to learn models with superior worst-group accuracy on the OOD test dataset without sacrificing average accuracy. To achieve this goal, a common surrogate optimization objective function that past literature has employed is to learn models to maximize worst-group accuracy on the OOD validation dataset (See Equation 2; $G_{val}$ denotes groups in the validation dataset).

$$\min_{\theta \in \Theta} \max_{g \in G_{val}} l_g(\theta). \tag{2}$$

This *minimax* approach to robust model selection is suboptimal since it ignores the predictive signal from other groups, i.e., in the optimization process it only upweights (or directly optimizes, as done by DRO or GroupDRO) the worst group. It also simplistically assumes that the worst-performing group on the validation dataset is distributionally similar to the worst group on the OOD test set.[2] So, instead of this *hard* minimax approach, we propose a *soft-minimax* approach that weights the errors from several poorly performing groups on the validation dataset to inform the hyperparameter choices for model selection. Intuitively, our approach can be seen as performing smoothing by borrowing statistical strength from several groups instead of just the worst group. We leverage the information retrieval and learning to rank literature to help us operationalize this soft group-weighting. This literature contains several ranking-based metrics that provide discounted importance to various items, e.g., the Discounted Cumulative Gain (DCG) metric.

### 4.1 Discounted Cumulative Gain (DCG) for Model selection:

First, use any base learner model, e.g., ERM to get the classification losses incurred by the different groups on the validation set. Next, sort all the $m_{val}$ groups according to their loss $g(1), g(2), \ldots, g(m_{val})$ from the largest (worst) to the smallest (best group). Then the composite DCG metric with a cutoff $k$ becomes,

$$\text{DCG@k}(\theta) = \sum_{i=1}^{k} \frac{l_{g(i)}(\theta)}{\log_2(i+1)}. \tag{3}$$

---

[2]The objective function that is optimized is based on the worst-group error which implicitly assumes that a similar error distribution would be present in the OOD test data. However, both validation and test datasets are OOD w.r.t training data, so this assumption could potentially be incorrect.

Equation ( 3) considers the $k$ groups with the highest OOD validation errors and provides them increasing weights (higher weight for worse performing group).

Since the $1/\log(x)$ function flattens fast as the number of groups increases, one can use DCG[3] at the quantile-level as opposed to group-level. The quantile-level DCG takes in a list of quantiles $\boldsymbol{q} = [q_1, \ldots, q_k]$ that corresponds to groups $g^{(q_1)}, \ldots, g^{(q_k)}$ at these quantiles, for example, $\boldsymbol{q} = [0, 1, \ldots, k]$ are the groups at quantile 0 (worst-group), quantile 1, up to quantile $k$. This leads to a slightly modified expression for DCG as shown below:

$$\text{DCG}_{\boldsymbol{q}}@k(\theta) = \sum_{i=1}^{k} \frac{l_{g^{(q_i)}}(\theta)}{\log_2(i+1)}. \tag{4}$$

### 4.1.1 Evaluation of the Model Selection metric

We just described a soft minimax-based model selection strategy, which generalizes the hard minimax used previously in the literature. Recall that the ultimate goal of effective model selection is to choose a model with superior performance on the OOD test set. So, how do we evaluate the effectiveness of our proposed metric over alternative model selection metrics?

An excellent way to think about this is how much concordance or agreement exists between the models selected by a given metric on the validation and the OOD test sets. A superior model selection metric should yield similar rankings of candidate models on either validation or test datasets. Thus, the best model on the validation set will also be the best model on the test data leading to effective model selection. For instance, consider our soft minimax metric; let's assume it ranks three candidate models as S2> S1> S3 based on validation set accuracy. Then, if the test set accuracies[4] of these three models are also S2> S1> S3, we consider the metric a good model selection strategy, and we can pick model S2 from this class of models. We use this intuition to guide our evaluation strategy for the model selection metric.

Let $\text{r}_{val}(M)$ denote the ranked accuracy list of models based on metric $M$, e.g., hard minimax, soft minimax, or average, on the validation set. Next, let $\text{r}_{test}(\text{worst-group-accuracy})$ represent a similar model accuracy list but based on worst group performance on the test set. Then, the metric $M$'s *concordance $C(M)$* can be defined as the similarity between $\text{r}_{val}(M)$ and $\text{r}_{test}(\text{worst-group-accuracy})$. Hence, a superior evaluation metric should have a high degree of similarity between the two rankings.

$$C(M) = \text{similarity}(\text{r}_{val}(M), \ r_{test}(\text{worst-group-accuracy})). \tag{5}$$

The *similarity* operator in Equation ( 5) can be operationalized by a function such as euclidean distance or cosine similarity.

### 4.2 Learning-To-Rank inspired novel method for Model Training

We just saw the use of DCG to *select* the best model from candidate models efficiently. Next, we propose a new method for *training* a model. Inspired by the recent success of the Just Train Twice (JTT) method (Liu et al., 2021) for group distributional robustness, we propose a method that performs iterative upweighting of training examples. However, unlike JTT, it leverages group annotations at training time. Our novel approach, called Discounted Rank Upweighting (DRU), iteratively ranks the groups by their accuracy and then upweights poorly performing groups. The key idea is to upweight training samples from the groups with the highest training errors and assign them differential importance commensurate with their ranking.

---

[3]It is important to note that the connection to DCG was inspired by learning-to-rank literature and was chosen to provide a familiar conceptual anchor for our readers. However, while our approach does draw certain conceptual parallels with DCG, it more accurately aligns with the principles of the Ordered Weighted Averaging (OWA) operator (Busa-Fekete et al., 2017; Do & Usunier, 2022; Yager & Kacprzyk, 2012), especially in terms of weighted aggregation. This alignment becomes particularly clear in the context of aggregating group losses, where the OWA's flexible framework for nuanced weighting schemes better represents our approach.

[4]Recall that on test data we only care about worst group accuracy.

At each epoch $t$ (excluding the first one) during the training process, a sample $x$ with label $y$ in the group $g \in \mathcal{G}_{train}$ is upweighted as

$$
w_g^t = \begin{cases} \frac{\log_2(C+2)}{\log_2(r_g^{t-1}+2)} & r_g^{t-1} \leq C \\ 1 & r_g^{t-1} > C \end{cases}
\tag{6}
$$

where $r_g^{t-1}$ is either the ranking index or the ranking quantile of the group $g$ in the training set (ascending order of training accuracy) evaluated from the previous (t-1) epoch. $C$ is a hyperparameter that controls the cutoff for upweighting (akin to $k$ in $DCG@k$). If the group ranking is greater than the cutoff, then the weight is one, that is, no upweighting. Otherwise, if the group has lower accuracy, then, it will be weighted by the discounted log function shown above. Note that the constant '2' in this function is used to have discounted factors for training groups that are consistent with those proposed by the standard DCG metric (Järvelin & Kekäläinen, 2002). If the upweighting is applied to all samples of each group regardless of their classification accuracy in the previous epoch, then the training objective of each epoch for a model with parameters $\theta$ is

$$
J_{DRU}^t(\theta) = \sum_{g \in \mathcal{G}_{train}} \sum_{(x,y) \in g} w_g^t * \mathcal{L}(x, y; f_\theta)
\tag{7}
$$

for $t \neq 0$. As one can infer, the first epoch is always the standard ERM training.

The upweighting scheme shown in Equation ( 7) upweights all the samples from a given group. One can also choose to upweight only the misclassified samples from the previous epoch. Assuming the misclassified samples to constitute an error set $E$, the modified training objective function becomes:

$$
J_{DRU}^t(\theta, E) = \sum_{g \in \mathcal{G}_{train}} \left[ \sum_{(x,y) \in g \cap E} w_g^t * \mathcal{L}(x, y; f_\theta) + \sum_{(x,y) \in g \setminus E} \mathcal{L}(x, y; f_\theta) \right]
\tag{8}
$$

We compare both these objective functions in our empirical results.

### 4.2.1 DRU Convergence

We can show the convergence of our method by drawing upon the methodology used by Xiao et al. (2023), who establish convergence for rank-based loss minimization. Our DRU objective is compatible with their framework, considering that we determine group ranks based on group loss values, thereby creating a sequence of ordered groups. The objective function is then defined as a weighted sum of these group losses, where the weights are calculated as the product of group size and reweighting factors assigned based on group rank.

When group sizes are uniform, our algorithm perfectly aligns with the rank-based loss minimization model described in Equation (2) of Xiao et al. (2023), indicating that convergence is achievable in such scenarios. Even for varying group sizes we can use the rank-based minimization framework by modifying our algorithm slightly—altering how we determine quantiles. Instead of grouping by quantiles, we use a cumulative counting method. For example, let's say we have three groups ranked 1, 2, and 3, with sizes of 10, 10, and 20, respectively. Here, the first group would cover the 1st to 25th quantile, the second the 25th to 50th, and the third the 50th to 100th. Our objective function then becomes the sum of average losses across these quantiles, where each is multiplied by specific weights (determined by DCG weighting). Groups that cover multiple quantiles have their weights calculated as a combined total of the weights for each quantile they span, proportionally adjusted based on the number of instances they include. This approach results in a consistent increase in average losses as we move to higher quantiles, making our objective consistent with Xiao et al. (2023)'s model. Hence, our procedure converges. This slightly modified DRU method based on counting cumulative examples up to a quantile gives slightly worse performance than the qDRU approach but performs better than ERM. We provide more detailed explanations in the Appendix.

# 5 Experimental Setup

## 5.1 Datasets

We use three real-world text classification datasets that have been commonly used in prior group robustness literature: AMAZON-WILDS (Koh et al., 2021), IMDB Movie Review Dataset (Pal et al., 2020), and a variation of Yelp Open Dataset[5]. The prediction task for all three datasets is to classify the review text into its corresponding 1-to-5 star rating. Each review is associated with a group, which corresponds to all reviews written by the same *user*. Each dataset has an in-distribution (ID) training set and out-of-distribution (OOD) validation and test sets. The OOD validation and test sets comprise reviews from disjoint sets of users (groups). The users in the training dataset are randomly split 50/50 to be in the ID validation and ID test datasets. Table 1 provides the summary statistics of each dataset. For more preprocessing and descriptive dataset details, please refer to the Appendix A. Intuitively, the performance of a ERM model should significantly downgrade on OOD validation and test sets compared to ID validation and test sets (Koenecke et al., 2020; Caldas et al., 2018). Table 1 confirms the significant accuracy drops from ID to OOD on all three datasets.

Table 1: Dataset Details. *Note:* 1) Number of groups is provided in the format (training, OOD validation, OOD test). 2) The ERM model accuracies are given in the format (average, 10-th percentile, worst group). All three performance metrics are lower on OOD val/test sets than on their ID counterparts. 3) 10-th percentile group is one that has a lower accuracy than 90% of all groups.

| Dataset | # Groups | Group size | ID val | OOD val | ID test | OOD test |
|---|---|---|---|---|---|---|
| AMAZON | (1252, 1334, 1334) | 75 | (75.7, 58.7, 24.0) | (72.3, 54.7, 6.3) | (74.7, 57.3, 24.0) | (71.9, 53.3, 12.0) |
| IMDB | (666, 561, 560) | 25 | (64.7, 46.7, 26.7) | (62.6, 43.1, 15.6) | (65.4, 48.0, 20.0) | (63.2, 42.9, 15.0) |
| Yelp | (500, 523, 522) | 100 | (65.2, 54.9, 41.0) | (64.5, 54.0, 34.0) | (64.0, 55.9, 26.9) | (63.0, 52.0, 18.0) |

We also generate several synthetic datasets in which each observation of a group is sampled from a 'shared' signal across all groups plus an 'idiosyncratic' signal which is group-dependent. The distribution and strength of the two signals are different across different datasets. Please refer to Section 7 for more details.

## 5.2 Experiment Setup for Model Selection

We consider the following metrics that one can use to select models/hyperparameters from a validation set: (1) **worst-group:** accuracy of the worst group; (2) **average:** average across all groups; (3) **10th percentile:** the accuracy of the group at the 10th percentile (lower accuracy than 90% groups); (4) **gDCG@k:** DCG at group level for the k percent worst-performing groups (k=10, 50); (5) **qDCG@k:** DCG at quantile level for percentiles [0,1,...,k] (k=10, 50).

To compare the effectiveness of these metrics for model selection, we trained DistilBERT base-learner models using JTT. We varied the two hyperparameters (first stage step $T$ and upweighting factor $\lambda$) of JTT to generate 16 candidate models. In particular, we considered $T \in \{1, 2, 3, 5\}$ and $\lambda \in \{2, 3, 5, 10\}$. Next, we rank these 16 models using the various metrics on their OOD validation set accuracy and then rank all the 16 models on worst group accuracy on the test OOD dataset. This process provides us with a ranked list of 16 models for each model selection metric on the validation set and another list ranking all the 16 models on their worst group accuracy on the OOD test set. Finally, we can assess the model selection performance of all the metrics by computing the similarity between their rankings of models on validation set with the "ground-truth" ranking of models on the test set (cf. Equation 5). In particular, we calculate the similarity of two ranked lists using euclidean distance, cosine similarity, and the NDCG of the model ranking on validation using the ranking on test as gold standard scores. The metric(s) which ranks the models on validation into the most similar positions with the true performance of the models on test set is(are) the most effective for selection.

---

[5]https://www.yelp.com/dataset/

### 5.3 Experiment Setup for Model Training

Similar to the DCG-based model selection, the DRU-based upweighting can also be performed by ranking the quantiles of the groups, i.e., **qDRU**, or simply ranking the indices of the groups, i.e., **gDRU**. **qDRU** is preferable when the number of groups is large since the logarithm function flattens out quickly in such a case. These upweighting methods also have **one** hyperparameter $C$, which controls the cutoff or the amount of smoothing.

Since we want to contrast with the hard minimax approach throughout this paper, we train models by only upweighting samples of the worst-performing group from the previous training epoch by a constant weight $\lambda$. It is easy to see that this hard minimax approach is a special case of our soft minimax approach in which the cutoff for DRU is the rank of the worst group (0). We denote this boundary case as **Worst** in our results.

In their basic form, our upweighting methods **qDRU**, **gDRU** upweight all the samples from certain groups (cf. Equation 7). A related upweighting strategy can be to further zoom in to each group and only upweight the misclassified examples from that group (cf. Equation 8). We experiment with this seemingly more precise weighting strategy and denote it using the suffix "+M" in our results. For clarity, the default strategy of upweighting all examples from a group is suffixed "+G". This leads to four different variants of our DRU models, **qDRU + M**, **qDRU + G**, **gDRU + M**, and **gDRU + G**.

We further compare against another variant of the upweighting strategy that upweights only the misclassified examples from the previous training epoch by a constant factor $\lambda$. We call this approach **Const** in our results. This variant will help us isolate the impact of the ranking-based logarithmic weighting since it is plausible that the improved accuracy might not be sensitive to the differential upweighting. Note that the **Const** method can not be applied to all the samples from each group (**+G**) since upweighting all the examples by the same amount makes the weights useless.

In addition to these methods, we compare against baseline methods **ERM**, **Group DRO**, and **JTT** described in Section §3.1. All methods except ERM and JTT have one hyperparameter ($C$ for DRU-based methods, Const, and Worst, and stepsize for Group GRO). We also considered **IRM** (Arjovsky et al., 2019) as a potential baseline but could not obtain comparable performance to other baselines (ERM, Group DRO, JTT). We hypothesize that IRM does not fit our scenario where there are a large number of groups. We therefore do not include IRM in the following experiments.

We follow the lead of the authors of the WILDS Distribution Shift Benchmark Suite (Koh et al., 2021) and use a finetuned base uncased DistilBERT model as our base learner in all our experiments. We used the following hyperparameters for DistilBERT as also suggested by (Koh et al., 2021): batch size 16; learning rate $1 \times 10^{-5}$ for AdamW optimizer (Loshchilov & Hutter, 2017); L2-regularization strength 0.01; 5 epochs with early stopping; and a maximum number of 512 tokens. Next, for both **qDRU** and **gDRU**, we performed a grid search to tune the cutoff hyperparameter $C \in [10, 20, 50, 100]$. $\lambda$ is selected from the list [2, 3, 5] for all methods that performed constant upweighting. For **JTT**, $T \in \{0, 1, 2\}$ and $\lambda \in \{2, 3, 5\}$. Finally, for **Group DRO** we fixed the step size as 0.01 following the best practice reported in (Koh et al., 2021). For the DistilBERT, we use the implementation of HuggingFace[6]. All the experiments are run on the NVIDIA GEFORCE RTX 2080 Ti using the PyTorch Framework. It is worth noting that DRU-based methods involve a single training pass with multiple epochs, similar to GroupDRO and ERM. This is in contrast to the JTT approach, which requires training the model with ERM twice, each involving multiple epochs. The biggest cost of DRU method is attributed to the computation of rank and subsequent reweighting. Since this ranking is determined based on the groups, the runtime complexity for the rank and reweight process can be estimated as $O(\|g\| \log \|g\|)$, where $\|g\|$ represents the number of groups. In practice, the training time ranged from 1.25 to 1.35 times that of ERM for the datasets used in this paper. Hence, our approach is highly computationally efficient.

---

[6]https://huggingface.co/docs/transformers/model_doc/distilbert

# 6 Results and Discussion

## 6.1 Model Selection Results

Table 2 shows the model selection results for Amazon, IMDB, and Yelp datasets. We report the similarity between the ranked list of the 16 candidate models on OOD validation set corresponding to each model selection metric and the ranked list of the models by their true accuracy on the worst group in OOD test.

As can be seen from the results, the "worst-group," that is, the *hard minimax* approach, usually performs worse compared to other metrics. This is surprising since it has identical semantics to the ground-truth metric we used on the test data (worst-group accuracy), confirming that the *hard minimax* metric has poor generalizability when distribution shift is present. The quantile-level DCG metrics (qDCGs) perform the best on all three datasets. Specifically, qDCG@10 performs the best on Amazon and IMDB, and qDCG@50 performs the best on the Yelp dataset.

Table 2: Concordance between the ranked lists of the models on OOD validation by different metrics and the ranked list by worst-group accuracies on OOD test. *Note:* ED = Euclidean Distance (lower is better), CS = Cosine Similarity (higher is better), NDCG = Normalized Discounted Cumulative Gain using test-worst-group ranking list as the gold standard (higher is better).

| Metric | Amazon | | | IMDB | | | Yelp | | |
|---|---|---|---|---|---|---|---|---|---|
| | ED | CS | NDCG | ED | CS | NDCG | ED | CS | NDCG |
| worst-group | 27.0 | .72 | .77 | 14.5 | .93 | .92 | 15.3 | .92 | .91 |
| average | 27.0 | .74 | .78 | 15.9 | .92 | .88 | 12.9 | .94 | .97 |
| 10th percentile | 23.5 | .80 | **.88** | 15.3 | .92 | .93 | 13.6 | .93 | .96 |
| gDCG@10 | 21.4 | .84 | .86 | **12.6** | **.95** | **.95** | 14.0 | .94 | .94 |
| gDCG@50 | 24.0 | .80 | .82 | 15.6 | .92 | .93 | 12.5 | .95 | **.97** |
| qDCG@10 | **20.4** | **.85** | .86 | **12.6** | **.95** | **.95** | 13.2 | .94 | .94 |
| qDCG@50 | 24.0 | .80 | .82 | 15.0 | .92 | .93 | **12.2** | **.95** | **.97** |

The "10th percentile" metric works better than worst-group or average accuracy, although not as good as the DCG-based metrics. This is understandable as the "10th percentile" worst-group is a special case of rank-based metric and it also smooths the *hard minimax* to a certain extent.

A hidden but practically important strength of the DCG-based metric that is not visible in the result tables is its ability to break ties between candidate models. In our experiments, 43.75% and 37.5% of models have identical 10th percentile and worst-group accuracies on all three datasets, respectively. This lack of model identification makes it hard to assess which model is the best, and one has to resort to suboptimal heuristics such as random tiebreaks to choose the best model. Ties are rare in the case of DCG-based metrics since they evaluate models using discounted (logarithm weighted) ranks of several poorly performing groups instead of just a single accuracy number as done by the "worst-group" or "10th percentile" metrics. Thus, the smoothing produced by our soft minimax metrics leads us to select better models. However, our results are still a bit inconclusive regarding how much smoothing is optimal (with regard to the cutoff threshold of DCG) since that threshold hyperparameter varies over the datasets in our experiments.

## 6.2 Model Training Results

The results of the test OOD datasets for the various methods are shown in Table 3. The tables report average and 10th percentile group accuracy for completeness, but the worst-group accuracy on the test OOD dataset is the target. Broadly, we see a trend that the DRU-based methods, which smoothly upweight multiple poorly-performing groups, outperform ERM and other methods with hard upweighting rules. Among DRU-based methods, **qDRU+M** provides the highest worst group accuracy (target metric) on the OOD test dataset on average, with a comfortable margin of improvement over **JTT**, **Group DRO**, **Worst**, **Const** which are statistically significant under a bootstrapped t-test. The 10th percentile and average group performance of **qDRU+M** and other DRU-based methods are also competitive or even better than the baselines,

suggesting that our soft-minimax-based methods improve OOD worst-group performance without a perceptible sacrifice of accuracy on better-performing groups or average accuracy. We also showcase similar improved performances of our methods on several synthetic scenarios (Section 7). In all these scenarios, DRU-based methods still significantly outperform the baseline methods, and **qDRU+M** is nearly consistently the best method on average, 10th and worst group accuracy metrics. We observe a slightly different pattern on real-world datasets where, interestingly, **qDRU+G** model consistently outperforms others on 10th percentile accuracy. Finally, we would like to note that real-world scenarios are often considerably more complex than the controlled synthetic setting since individual groups and misclassified examples may be more affected by random noises or even adversarial signals (e.g., spam or fake reviews). And in these scenarios, the 10th percentile group accuracy may be a more reasonable target for group robustness, and upweighting all examples in a group may be more resilient to noise than only upweighting misclassified examples. To summarize the three metrics into one for easier comparison, we also report the t-statistics of group accuracy of each method in Appendix C, which shows our methods did better in reducing the variance among groups without sacrificing the average group performance.

Table 3: Results on OOD Test. G: upweighting all samples of a group (Equation 7). M: upweighting misclassified samples of a group (Equation 8). qDRU: upweighting according to ranking percentile. gDRU: upweighting according to ranking index. The results are in the format of (average accuracy/10th percentile accuracy/worst group accuracy). Bold: best OOD test performance (bootstrapped t-test)

| Dataset | ERM | Group DRO | JTT | Worst + G | Worst + M |
|---|---|---|---|---|---|
| Amazon-WILDs | **71.9**/53.3/12.0 | 70.0/53.3/8.0 | 71.6/53.3/9.3 | 72.2/53.3/12.0 | 70.2/53.3/17.3 |
| Yelp | **63.0**/52.0/18.0 | 59.2/49.0/**27.0** | 61.7/51.0/19.0 | 62.8/52.0/20.0 | 62.7/**53.0**/25.0 |
| IMDB | 63.2/42.9/15.0 | 61.0/42.0/10.8 | 62.5/42.3/10.0 | 62.5/42.9/22.5 | 63.3/43.8/17.5 |
| Dataset | qDRU + G | qDRU + M | gDRU + G | gDRU + M | Const + M |
| Amazon-WILDs | 70.2/**54.7**/17.3 | 70.1/53.3/**18.7** | 70.2/53.3/17.3 | 71.5/**54.7**/14.7 | 71.0/53.3/14.7 |
| Yelp | 62.6/**53.0**/23.0 | 62.5/52.0/21.0 | 62.8/**53.0**/24.0 | 62.1/52.0/**27.0** | 63.0/**53.0**/21.0 |
| IMDB | **64.1**/**45.8**/20.0 | 62.0/43.3/**25.0** | 61.0/42.9/18.9 | 62.4/43.8/15.0 | 62.6/44.1/15.0 |

# 7 Synthetic Data Experiments

We also showcase our method's improved performance in a controlled environment where we generate the data using a fixed data-generating process. Past work has also used synthetic data to validate new methods for distribution shift (Arora et al., 2021; Arjovsky et al., 2019).

## 7.1 Synthetic Data Generation

The procedure for synthetic data generation is summarized in Algorithm 1 (in Appendix). The main goal of this algorithm is to create different distribution shifts across groups by differently mixing two predictive signals for each observation of a group. The first signal is a "shared signal" present across all the groups and easily captured by any model and the second signal is an "idiosyncratic signal" that varies significantly across groups. As part of our controlled simulation setup, we vary the percentage of groups $U$ containing idiosyncratic signals in a dataset. Specifically, we model the shared signal $M$ via a Gaussian distribution $\mathcal{N}(\mu_s, \sigma_s)$ and the idiosyncratic signal is operationalized by a set of $W$ Gaussian distributions whose each element represents a unique idiosyncratic signal. For each sample $i$ of a given group $g$, we sample a shared signal $\mathbf{S}_g^i$ from the distribution $M$. Next, if the group $g$ contains idiosyncratic signals (sampled using Bernoulli($U$)), one idiosyncratic signal distribution $w_g$ is sampled from $W$ based on a prior distribution $p$. Then for each sample $i$ of the group, an idiosyncratic signal $\mathbf{D}_g^i$ is sampled from $w_g$. Then, another simulation parameter $a \in [0, 1]$ that controls the strength of the idiosyncratic signal is sampled from a truncated Gaussian distribution. Note that for a group without idiosyncratic signals, each sample of the group will only have the shared signal. Finally, we get the feature representation for a given sample $i$ of the group $g$ as $\mathbf{x}_g^i = \mathbf{S}_g^i + \text{Bernoulli}(U) \times a \times \mathbf{D}_g^i$. The label of the sample is obtained as $y = \mathbb{1}_{\mathbb{R}_+}(f(x_g^i) + \mathcal{N}(0, \sigma^2))$,

where $\mathbb{1}_{\mathbb{R}_+}$ is the indicator function, $f$ is a function that transforms features into labels (e.g., sin function) and $\sigma^2$ is the variance of a random Gaussian noise added to every instance.

## 7.2 Experiment setup for Synthetic Data

As shown in Algorithm 1 (in Appendix), in our experiments, the dimension of all signals is 2. The shared predictive signal $M$ is $\mathcal{N}([0, 0], 4I)$ where $I$ is the two-dimensional identity matrix. There are four idiosyncratic signals in $W$ and their mean and variance values were chosen as $[([0.25, 0.25], I), ([0.25, -0.25], I), ([-0.25, 0.25], I),$ and $([-0.25, -0.25], I)]$ respectively. $f$ is a sine function that takes the sum of all feature dimensions as input. The strength factor $a$ is sampled from a truncated Gaussian distribution $\mathcal{N}(0.75, 0.25, 0, 1)$ and $\sigma = 0.5$ for the random noise. We also evaluate a setting with larger noise levels, using variances ($\sigma^2$) of 1 and 4. The results are shown in the Appendix F.

Using these parameters, we generate synthetic datasets under four different settings as shown in Table 7 (in Appendix). Setting 1 is the one that exhibits the slightest distribution shift since 80% of the groups in the training, validation, and test sets contain the same set of idiosyncratic signals. Setting 2 shows a realistic real-world scenario where the training data uniformly ($p_{train}(w_i) = 1$) contains each of the idiosyncratic signals, but only 20% of the training groups have an idiosyncratic signal. The test data, on the other hand, contains the idiosyncratic signal in 80% of the groups. The third and fourth settings show substantial distribution shifts since they represent the case where some of the idiosyncratic signals are altogether hidden from the training dataset. This happens routinely in real-world scenarios when the training dataset is not large enough to include all the unique signals introduced by unseen groups in the OOD test data.

We generate 1000 training groups, 500 test OOD groups, and 500 validation OOD groups for each of these settings. Each group contains 75 samples. The base learner for training all these datasets is a three-layer feed-forward neural network with a hidden state size of 128. Each layer is connected by LeakyReLU (Xu et al., 2015), and a 0.5 dropout rate is applied. We performed a grid search $C \in [5, 10, 20, 50, 100]$ to select the best cutoff for **qDRU**. For all the upweighting methods with constant factors, i.e., **Worst**, **Const**, and **JTT**, $\lambda$ is selected from the list $[2, 3, 4, 5]$. The step size for **Group DRO** is chosen as 0.01, and the first and second training steps of JTT were 5. Finally, we performed the model selection using **qDCG@10** metric, which was the best performing metric as we saw in Section 6.1.

## 7.3 Synthetic Data Results

The results are shown in Table 4, and as can be seen, the DRU-based methods significantly outperform the baseline methods in all simulation settings. DRU variants significantly boost the worst group accuracy (the last number in the accuracy lists (90,70,50) in Table 4) by up to 10% in some cases compared to the baselines. Overall, **qDRU+M** is consistently the best DRU-variant except for Setting 1, in which there is only a mild distribution shift. Group DRO and JTT also perform as well as DRU-based methods in Setting 1, but their relative performance drops in settings with significant distribution shifts. Interestingly, the constant upweighting method **Const+M** performs even worse than **ERM** which doesn't perform any weighting at all. When significant distribution shifts are present (especially in setting 3 and 4 where unseen idiosyncratic signals are present in the test dataset), **qDRU+M** not only improves the OOD worst group accuracy but also the 10th percentile of worst groups, and it even shows a significant improvement (over 6%) in *average* test accuracy compared to ERM, GroupDRO, and JTT baselines. The performance of **qDRU+G** is notable in Setting 4 where most significant distribution shift is present, and its performance is only inferior to **qDRU+M**.

# 8 Conclusion and Future Work

In conclusion, this paper highlights the weakness of the canonical approach in group distributional robustness literature of focusing only on the worst group accuracy for model selection and model training. We introduced a suite of methods inspired by the information retrieval and learning-to-rank literature for group robust model selection and model training. Essentially, our ranking-based soft minimax approaches smooth the predictive signal learned at training time by performing a discounted weighting which leads to improved

Table 4: Results on Synthetic Dataset in the format of (average accuracy/10th percentile accuracy/worst group accuracy). **BOLD**: best OOD test performance under the same setting. Underline: second best performance.

| Dataset | ERM | Group DRO | JTT | Worst + G | Worst + M |
|---|---|---|---|---|---|
| Setting 1 | (70.2/62.7/54.7) | (**78.7**/**72.0**/62.7) | (76.5/69.3/62.7) | (76.6/69.3/58.7) | (77.8/**72.0**/62.7) |
| Setting 2 | (68.1/61.3/52.0) | (76.0/69.3/57.3) | (73.1/66.7/58.7) | (78.1/72.0/62.7) | (76.9/70.7/58.7) |
| Setting 3 | (68.9/61.3/50.7) | (75.7/69.3/60.0) | (76.3/70.7/61.3) | (77.4/70.7/64.0) | (76.3/69.3/61.3) |
| Setting 4 | (66.9/60.0/50.7) | (75.3/68.0/60.0) | (75.2/69.3/60.0) | (76.5/69.3/61.3) | (76.7/70.7/62.7) |
| **Dataset** | **qDRU + G** | **qDRU + M** | **gDRU + G** | **gDRU + M** | **Const + M** |
| Setting 1 | (77.2/70.7/61.3) | (77.6/70.7/64.0) | (75.9/69.3/60.0) | (78.5/**72.0**/**65.3**) | (67.0/60.0/45.3) |
| Setting 2 | (77.7/72.0/61.3) | (79.0/**73.3**/**65.3**) | (77.3/70.7/58.7) | (**79.5**/**73.3**/**65.3**) | (67.6/61.3/52.0) |
| Setting 3 | (77.1/70.7/62.7) | (**81.0**/**74.7**/**69.3**) | (76.8/69.3/60.0) | (76.4/69.3/62.7) | (70.3/64.0/56.0) |
| Setting 4 | (79.4/73.3/65.3) | (**80.4**/**74.7**/**68.0**) | (77.3/70.7/60.0) | (76.9/70.7/62.7) | (66.9/60.0/49.3) |

generalization performance on the OOD test dataset in the challenging *domain generalization* (Koh et al., 2021) setting. Our theoretical intuition regarding the fit of ranking-based methods for group robustness is backed by our methods' equally strong empirical performance on synthetic and several real-world benchmark datasets. Group identities carry a strong predictive signal (even if they do not overlap in training/test) since we observe that group-based approaches perform better than those that ignore the group structure. Though, more research needs to be done to investigate this deeply. The learning to rank literature implicitly assumes orthogonality between the search results (or group features in our case). So, as part of future work, it will be interesting to study how the number of groups and their correlation structure impacts the performance of the ranking-based methods.

## Broader Impact Statement

We did not notice any immediate ethical issues in our work. Our proposed methods improve the tail-group accuracy, so our approach ensures that we do not adversely impact marginalized and disadvantaged groups. For the licenses, the Amazon-WILDS dataset (Koh et al., 2021) is licensed under the MIT license. The IMDB dataset (Pal et al., 2020) is licensed under CC-BY 4.0. The Yelp Open Dataset provides a YELP DATASET TERMS OF USE with permission to use for academic purposes. Our use of the three existing datasets for academic purposes is consistent with their intended use. All usernames in these datasets are anonymized into hashing values.

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

## A  Dataset Descriptions

**1) AMAZON-WILDS:**  Collected as part of the WILDS dataset suite (Koh et al., 2021), the Amazon-WILDS dataset involves predicting star ratings from users' reviews of Amazon products. The training set has $245,502$ reviews from 1252 users (at least 75 reviews per user). The ID validation set consists of 46,950 reviews from 626 of the 1252 users in the training set. The ID test set is the same size as the ID validation set and contains reviews from the remaining 626 users from the training dataset. Finally, the OOD validation and the OOD test sets each have $100,050$ reviews (75 reviews per user) from $1,334$ new users.

**2) IMDB Movie Reviews:**  We downloaded the IMDB dataset from (Pal et al., 2020) and modified it to exhibit considerable OOD performance drops on the validation and test sets. To construct our dataset, we aggregate the data at the user level and split it into training, validation, and test sets using K-means clustering to ensure a significant distributional shift from ID to OOD sets. Specifically, we calculate the average of pre-trained DistilBERT embeddings of each user's reviews and then cluster their embeddings (k=2). One cluster is randomly selected as the ID set, and the other is the OOD set. Next, users in the OOD set are randomly split into OOD validation and OOD test sets, and the users in the ID set with at least 50 reviews are randomly divided into ID validation and ID test sets. The final training set has $41,146$ reviews from 666 users. The ID validation set consists of 20,070 reviews from 333 of the 666 users in the training set. Similarly, the ID test set contains 21,083 reviews from the other half of the users in the training dataset. The OOD validation and test sets include $42,703$ and $43,451$ reviews from 561 and 560 unseen users, respectively, with each user containing at least 25 reviews.

**3) Yelp Business Reviews:**  WILDS dataset suite (Koh et al., 2021) contains a modified version of the Yelp Open Dataset; however, there's no accuracy drop from their ID set to OOD set. Thus, we modify it by clustering at the user level in a similar fashion as we did for the IMDB dataset. We set k=6 and select the two farthest clusters as the OOD and ID sets to have a significant out-of-distribution performance drop. The training set comprises $64,931$ reviews from 500 users. The ID validation and test sets consist of $20,070$ and $21,083$ reviews from 333 out of the 666 users in the training set, respectively. Finally, the OOD validation and test sets include $52,200$ and $52,300$ reviews from 522 and 523 unseen users, respectively.

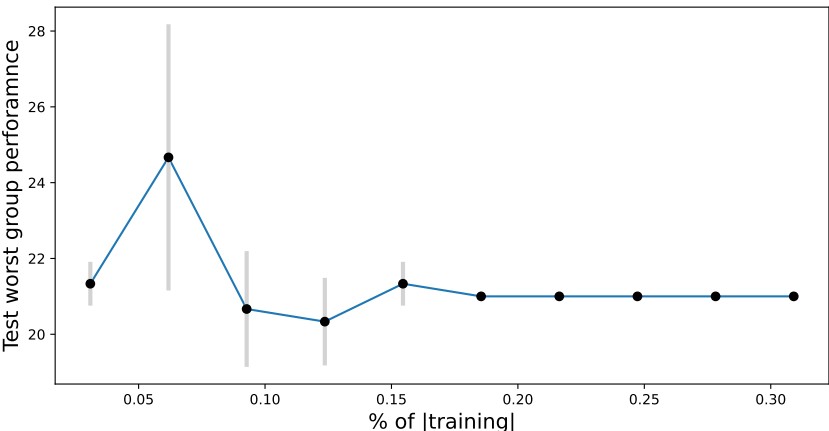

Figure 1: The worst group performance of the test data over the relative size of the validation data to the size of the training data in the Yelp dataset.

## B   Size of Validation Data

We evaluated the impact of validation data size on the model selection performance using the qDRU+M method. This was achieved by subsampling the original Yelp validation data in varying proportions, specifically $[10\%, 20\%, 30\%, \ldots, 100\%]$. For each proportion, we performed the subsampling process three times using different random seeds and then chose the best model based on each subsampled validation dataset. The performance of these selected models on the test data, particularly the worst group performance relative to the size of the training data, is depicted in Figure 1. Our results show that when the size of the validation data is more than 10% of the training data, the model performance demonstrates minimal degradation. However, with smaller validation datasets, we observed greater fluctuations in model performance, even though occasionally these models achieved superior results. This finding indicates that for effective and robust model selection using our methods, a validation set constituting at least 15% of the training data size (containing group annotations), is advisable.

## C   T-statistic of Real World Datasets

The t-statistics of each method on OOD datasets are shown in Table 5. For each method, the t-statistic is computed by dividing the average group performance by the standard error of all groups' performances. A robust model aims to have a higher worst group performance while not sacrificing the average performance which results in a smaller standard error of the groups' performances, thus a higher t-stat corresponds to a better model. The results show that our proposed methods are usually consistently better than ERM, JTT and GroupDRO.

Table 5: T-stats of group performance on OOD Test (T-stats here calculated as $\frac{\bar{p}}{SE(p)}$ where $\bar{p}$ is the average group performance, $SE(p)$ is the standard error of all groups' performance). A higher t-stat is achieved by reducing the variance among group performance (e.g., improving worst-group performance) while maintaining a high average accuracy. G: upweighting all samples of a group. M: upweighting misclassified samples of a group. qDRU: upweighting according to ranking percentile. gDRU: upweighting according to ranking index. Bold: best OOD test performance under the same setting.

| Dataset | ERM | Group DRO | JTT | Worst + G | Worst + M |
|---|---|---|---|---|---|
| Amazon-WILDs | 183.0 | 164.0 | 171.3 | 179.7 | 186.3 |
| Yelp | 164.0 | 169.2 | 165.1 | 169.2 | 176.1 |
| IMDB | 93.6 | 94.7 | 92.4 | 94.0 | 96.6 |

| Dataset | qDRU + G | qDRU + M | gDRU + G | gDRU + M | Const + M |
|---|---|---|---|---|---|
| Amazon-WILDs | 190.7 | **194.3** | 188.8 | 188.5 | 187.0 |
| Yelp | 173.3 | 169.2 | **177.7** | 165.6 | 172.0 |
| IMDB | **102.1** | 101.4 | 94.7 | 99.5 | 95.9 |

## D   Validation Performances of Real World Datasets

Results of validation sets of each method are shown in Table 6.

## E   Synthetic Data Generation Algorithm

The algorithm is shown in Algorithm 1.

## F   Synthetic Data With Different Noise Levels

We reported the performance of the qDRU + M model at noise variances $\sigma = 1$ and $\sigma = 2$ in Setting 2. The cutoff is fixed at 10 without additional fine-tuning. Table 8 illustrates that overall performance deteriorates

Table 6: Results on OOD Validation. G: upweighting all samples of a group. M: upweighting misclassified samples of a group. qDRU: upweighting according to ranking percentile. gDRU: upweighting according to ranking index. The results are in the format of (average accuracy/10th percentile accuracy/worst group accuracy). Bold: best OOD validation performance under the same setting.

| Dataset | ERM | Group DRO | JTT | Worst + G | Worst + M |
|---|---|---|---|---|---|
| Amazon-WILDs | (72.3/54.7/5.3) | (70.7/54.7/5.8) | (72.5/53.3/5.3) | (**72.9**/54.7/6.7) | (71.0/53.3/**8.0**) |
| Yelp | (**64.5**/54.0/34.0) | (60.5/51.0/31.0) | (63.2/52.0/32.0) | (64.0/54.0/35.0) | (64.2/54.0/35.0) |
| IMDB | (62.6/43.1/15.6) | (61.1/40.5/15.5) | (62.0/43.0/15.4) | (61.5/41.9/17.6) | (63.0/44.4/20.0) |

| Dataset | qDRU + G | qDRU + M | gDRU + G | gDRU + M | Const + M |
|---|---|---|---|---|---|
| Amazon-WILDs | (70.9/54.7/6.7) | (71.1/54.7/6.7) | (70.9/54.7/6.7) | (72.1/**56.0**/**8.0**) | (71.8/54.7/**8.0**) |
| Yelp | (64.0/53.1/**40.0**) | (64.3/**54.3**/**40.0**) | (63.9/54.0/34.0) | (63.6/54.0/38.0) | (64.4/53.0/37.0) |
| IMDB | (**63.3**/44.4/19.3) | (61.9/**46.0**/15.7) | (61.0/44.0/15.4) | (61.9/43.2/**20.2**) | (62.5/44.4/17.6) |

---

**Algorithm 1** Synthetic Data Generation

---

**Require:** $Q$: number of reviews per group; $U$: % of groups with idiosyncratic predictive signal, $W$: a set of predefined Gaussian idiosyncratic predictive signals; $p$: probability of each idiosyncratic signal; $M \sim N(\mu_s, \sigma_s^2)$, the shared predictive signal.

$\mathcal{S} \leftarrow \{\}$
**for** $g \in \mathcal{G}$ **do**
    $a_g \leftarrow Truncated\mathcal{N}(0.75, 0.25, 0, 1)$;
    $w_g \sim N(\mu_g, \sigma_g^2) \leftarrow random.choices(W, p)$;
    $has\_idiosyncratic \leftarrow \mathcal{U}(0, 1) \leq U$
    **for** $i = 1, 2, \ldots, Q$ **do**
        $shared_g^i \leftarrow \mathcal{N}(\mu_s, \sigma_s^2)$
        $x_g^i \leftarrow shared_g^i$
        **if** $has\_idiosyncratic$ **then**
            $idiosyncratic_g^i \leftarrow N(\mu_g, \sigma_g^2)$
            $x_g^i \leftarrow x_g^i + a_g * idiosyncratic_g^i$
        **end if**
        $y_g^i = \mathbb{1}_{\mathbb{R}_+}(f(x_g^i) + \mathcal{N}(0, 0.25))$
        $\mathcal{S} \leftarrow \mathcal{S} \cup \{(x_g^i, y_g^i)\}$
    **end for**
**end for**
return $\mathcal{S}$

---

Table 7: Four different synthetic dataset settings. $p$ is the prior distribution of the four idiosyncratic signals (before normalizing). $U$ is the portion of groups that have idiosyncratic signals in format of (training, validation, test)

| | $p_{train}$ | $p_{val}$ | $p_{test}$ | $U$ |
|---|---|---|---|---|
| 1 | (1, 1, 1, 1) | (1, 1, 1, 1) | (1, 5, 1, 5) | (0.8, 0.8, 0.8) |
| 2 | (1, 1, 1, 1) | (1, 1, 1, 1) | (1, 5, 1, 5) | (0.2, 0.2, 0.8) |
| 3 | (0, 1, 1, 1) | (1, 1, 1, 1) | (1, 5, 1, 5) | (0.2, 0.2, 0.8) |
| 4 | (1, 1, 0, 0) | (1, 1, 0, 0) | (0, 0, 1, 1) | (0.2, 0.2, 0.8) |

with increasing noise variance. However, our method demonstrates robust improvements in average, 10th percentile, and worst group metrics.

Table 8: Average, 10th and worst group performance of ERM and qDRU in different values of $\sigma$ in Setting 2.

| $\sigma$ | 0.5 | 1 | 2 |
|---|---|---|---|
| ERM | 68.9/61.3/50.7 | 66.3/58.7/49.3 | 57.8/50.7/38.7 |
| qDRU + M | **81.0/74.7/69.3** | **68.4/61.3/54.7** | **59.3/52.0/45.3** |

## G    DRU Convergence details

For our proposed algorithm DRU, the objective is

$$\min_{\boldsymbol{\theta}} \mathcal{L}(\boldsymbol{\theta}) = \sum_{i=1}^{N} w_i n_{[i]} Y_{[i]},$$

where the $\boldsymbol{\theta}$ denotes the model's parameters, and $N$ is the total number of groups in the training set. $Y_i$ represents the average training loss of the $i^{th}$ group for $i \in \{1, ..., n\}$. Let $Y_{[1]} \leq ... \leq Y_{[n]}$ be the order statistics. $w_i$ is the weight corresponding to the $i^{th}$ smallest group and $n_{[i]}$ is the number of instances within group $g_{[i]}$.

When group sizes are the same $n_{[1]} = n_{[2]} = \ldots = n_{[N]} = c$, we have

$$\mathcal{L}(\boldsymbol{\theta}) = c \left( \sum_{i=1}^{N} w_i Y_{[i]} \right) \tag{9}$$

whose convergence has been proved by Xiao et al. (2023) under the alternating direction multiplier method (ADMM) optimization.

However, when the group sizes differ, the proof of convergence is not guaranteed. The main challenge comes from the fact that $w_i * n_{[i]}$ varies with group size since $n_{[i]}$ (the number of instances in a group) depends on $g_{[i]}$. To address this, we employ a simple modification of our approach: Rather than assigning weights to each group, we split the dataset into the same-size quantiles and then assign the weights to each quantile accordingly. This can be better understood through Figure 2, where the groups are arranged in order of their group loss. In this arrangement, $g_{[N]}$ represents the group with the poorest performance, followed by $g_{[N-1]}$ as the second poorest, and so forth. Each triangle represents all instances of the corresponding groups. As the number of instances in each quantile ($Q_i$) is the same and groups may have different sizes, instances of some groups will be split into different quantiles (e.g. black and yellow part for $g_{[N]}$). The instances that fall in the same quantile (with the same color shown in Figure 2) will be upweighted by the same factor $w_q$ according to Equation 6. Then the new objective should be:

$$\mathcal{L}(\boldsymbol{\theta}) = n \left( \sum_{q=1}^{100} w_q \tilde{Y}_q \right) \tag{10}$$

where $n = \frac{\sum_{i=1}^{N} n_i}{100}$ is the number of instances in each quantile which is a constant. $\tilde{Y}_q$ is the average loss of quantile $q$. Then as long as the $\tilde{Y}_q$ is sorted, the convergence can be reached again according to Xiao et al. (2023) under ADMM optimization. This is feasible when we keep the losses of each part of the group across boundaries the same (e.g. if the black and yellow parts for group $g_{[N]}$ shown in Figure 2 have the same loss and so on for each group then $\tilde{Y}_q$ is sorted.) Given a set of instance losses of a group, deciding whether it can be partitioned into two (or multiple) subsets such that the sum of the numbers in each set is the same is naturally a partition problem (which is NP-Hard).

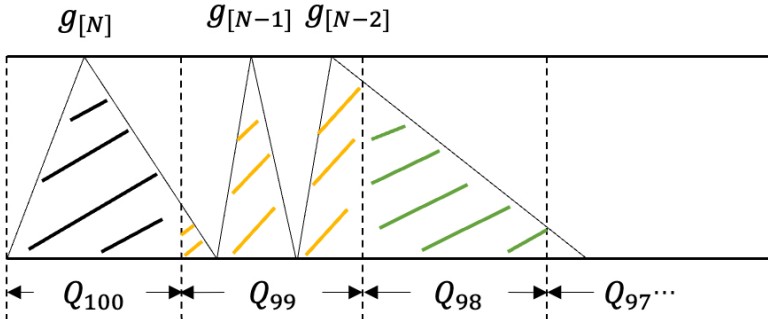

Figure 2: Figure showing division of instances into quantiles in the scenario in which groups differ in size. Suppose groups are ranked based on their loss, with $g_{[N]}$ representing the group with the worst performance and $g_{[N-1]}$ as the second worst, continuing in this order. Further, each triangle represents all instances belonging to its respective group. Given that each quantile ($Q_i$) contains an equal number of instances and the groups vary in size, instances from the same group may be distributed across different quantiles. This is illustrated by the division of $g_{[N]}$ into both black and yellow segments. Instances within the same quantile, even if they originate from different groups, are assigned the same upweighting factor which is determined by the DCG function (as visually represented by the consistent color coding in the figure.)

We side-step solving this intractable problem by assigning weights at the group level by taking a weighted average over the reweighting factors of parts of the group across boundaries.

For example, assume there are $x$ instances in $g_{[N]}$ that fall into $Q_{100}$(black) and $y$ instances that fall into $Q_{99}$(yellow), then the reweighting factor for the group is $\frac{x*w_{100}+y*w_{99}}{x+y}$. Generally, if a group $g_{[i]}$ spans across quantiles $Q_k, Q_{k-1}, \ldots, Q_{k-j}$, and the number of instances in each quantile is $n_{Q_k}, \ldots, n_{Q_{k-j}}$, then the weight for $g_{[i]}$ will be:

$$w_i' = \frac{\sum_{m=0}^{j} w_{k-m} n_{Q_{k-m}}}{n_{[i]}} \tag{11}$$

where $\sum_{m=0}^{j} n_{Q_{k-m}} = n_{[i]}$ the total number of instances of the group. And for any $m \neq 0$ and $m \neq j$, $n_{Q_{k-m}} = n$ should be the number of instances in each quantile. Even though knowing the real split of the group across quantiles is NP-hard, $n_{Q_k}, \ldots, n_{Q_{k-j}}$ is known since the size of the group and the number of instances in a quantile is known. (e.g., In Figure 2, if we know the number of instances in a quantile $n = 1000$ and further assume the group size of $g_{[N]}$ is $n_{[N]} = 1050$, then there should be 1000 instances in the black part of $g_{[N]}$ and 50 instances in the yellow part of $g_{[N]}$ and so on for all the groups.) Then, the new objective function can be written as:

$$\tilde{\mathcal{L}}(\boldsymbol{\theta}) = \left( \sum_{i=1}^{N} w_i' n_{[i]} Y_{[i]} \right) \tag{12}$$

We can prove that the reweighted loss of a group under objective 10 is the same as the reweighted loss of a group under objective 12. Without loss of generality, we can assume that a group $g_{[i]}$ should be split into two quantiles under the ideal objective 10 as $g_{[N]}$ in Figure 2. Assume the sets of instances in two quantiles are $\mathcal{X}$ and $\mathcal{T}$ correspondingly. The losses of instances in $\mathcal{X}$ are $\{X_1, ..., X_\alpha\}$ and the losses of instances in $\mathcal{T}$ are $\{T_1, ..., T_\beta\}$ where $\alpha$ and $\beta$ are the sizes of two parts and $\alpha + \beta = n_{[i]}$. Then under the ideal split so that each part has the same average loss, we have $\frac{\sum_{i=1}^{\alpha} X_i}{\alpha} = \frac{\sum_{j=1}^{\beta} T_j}{\beta}$. The average loss of the group

$Y_{[i]} = \frac{(\sum_{i=1}^{\alpha} X_i + \sum_{j=1}^{\beta} T_j)}{n_{[i]}}$. Under the objective 10, the reweighted loss of this group is then

$$
\begin{aligned}
&w_\alpha \sum_{i=1}^{\alpha} X_i + w_\beta \sum_{j=1}^{\beta} T_j \\
=&\frac{1}{\alpha}(\alpha w_\alpha \sum_{i=1}^{\alpha} X_i + \alpha w_\beta \sum_{j=1}^{\beta} T_j) \\
=&\frac{1}{\alpha}(\alpha w_\alpha \sum_{i=1}^{\alpha} X_i + \beta w_\beta \sum_{i=1}^{\alpha} X_i) \\
=&\frac{\sum_{i=1}^{\alpha} X_i}{\alpha}(\alpha w_\alpha + \beta w_\beta) \\
=&\frac{(\alpha + \beta) \sum_{i=1}^{\alpha} X_i}{(\alpha + \beta)\alpha}(\alpha w_\alpha + \beta w_\beta) \\
=&\frac{(\alpha w_\alpha + \beta w_\beta)}{\alpha + \beta}(1 + \frac{\beta}{\alpha}) \sum_{i=1}^{\alpha} X_i \\
=&w_g'(1 + \frac{\beta}{\alpha}) \sum_{i=1}^{\alpha} X_i \text{ according to Equation 11} \\
=&w_g'(\sum_{i=1}^{\alpha} X_i + \sum_{j=1}^{\beta} T_j) \text{ according to the even loss split} \\
=&w_g' n_{[i]} Y_{[i]}
\end{aligned}
\tag{13}
$$

We have shown that the reweighted loss of a group in objective 10 is the same as the reweighted loss of this group in objective 12 using the weighted reweighting factors in Equation 11. When a group is split into more than two quantiles, the proof can be applied recursively to adjacent quantiles as illustrated above. Hence, one can implement Objective 12 in practice without solving the NP-hard partition problem. Again, as mentioned earlier, the convergence of objective 10 has been proven by Xiao et al. (2023) under ADMM optimization.

