# OpenReview forum: "Group Robustness via Discounted Rank Upweighting"
_TMLR — Rejected by TMLR_

### Review · Reviewer_re2Y · 2023-10-30

**Summary Of Contributions:**

This paper proposes a new group distributional robust optimization algorithm with different weights for each group, with the conjecture that this will help mitigate the impact of spurious features and improve the ability to generate test data. The authors introduce the discounted cumulative gain metric from the learning-to-rank literature to determine the weights for each group. Experiments demonstrate the effectiveness of the proposed method.

**Audience:**

Yes

**Broader Impact Concerns:**

No further concerns.

**Claims And Evidence:**

Yes

**Requested Changes:**

See the weakness part.

In the experiments, it is suggested to add ablation studies on the number of validation data and computational efficiency comparison with baseline algorithms.

**Strengths And Weaknesses:**

Strengths:
1. This paper studies an important problem with a reasonable motivation, that is, compared to optimizing only the performance on the worst group, optimizing the weighted average accuracy over all groups can lead to more robust results in unseen test data.

2. The introduction of the discounted cumulative gain metric from the learning-to-rank literature to determine the weights for each group in Group DRO is novel to me.

3. Experiments on synthetic data and several benchmark datasets show the effectiveness of the proposed algorithm.

Weaknesses:
1. Although the empirical studies show good results, there is still a lack of theoretical guarantess and formal intuition discussion for the weight determination. It is hard to say under what condition the performance of the proposed soft-minimax strategy will be guaranteed.

2. Additional validation data set is required in the weight determination. This may limit the potential application of the proposed method. If the validation data set is small, the weights may be overfitted to these data.

3. The efficiency of the proposed method is questionable. There is an iterative learning process of the proposed algorithm, but there is no empirical comparison of computational efficiency. For example, the JTT methods require ERM only twice, but the proposed algorithm require multiple, this time consuming requires further discussion.

---

> ### Author Response · Authors · 2023-11-23
> **Thank You. Clarifications and Revisions**
>
> Thank you for the insightful review and for appreciating the key contributions and strengths of our paper. We will add the ablation studies where we vary the amount of validation data to the paper; we will also discuss the computational efficiency of the DRU approach, which as we discuss below is not inferior to JTT.
>
> Our detailed response is below:
>
> 1. Lack of Theoretical Guarantees: (We copy pasted our common response to all the the reviewers here, since we had enough space left)
>
> We appreciate the suggestion to include a formal theoretical analysis, as it would indeed deepen the understanding of our proposed algorithm. Our work primarily aims to introduce and empirically validate a novel algorithm tailored for real-world applications, particularly in scenarios where group distributional shifts are present. This focus aligns with similar studies in the field, notably the Just-train-twice (JTT) paper (ICML 2021), which has been a significant inspiration for our approach. While we acknowledge the value that a theoretical analysis would add, it's important to note that the absence of such analysis does not diminish the practical applicability of our algorithm. This perspective is exemplified by the JTT paper, which has demonstrated substantial practical value despite the absence of formal theoretical results. Nevertheless, we agree that further theoretical exploration could be a beneficial direction for future work and will consider this in our ongoing research.
>
> In DRU, reweighting is performed every epoch based on the group loss observed during that epoch. This means the objective function of the algorithm changes over epochs because the group weights are dynamically adjusted based on each group’s loss, which increases the complexity of analyzing the optimization procedure precisely. That said, there is theoretical work that can be related to the convergence of our proposed algorithm, in particular the methodologies outlined in recent papers [1,2]. Specifically, Paper [1] establishes the convergence of a reweighting algorithm that adjusts individual sample weights based on gradients. Paper [2] introduces a comprehensive framework for rank-based loss minimization and discusses strategies to achieve convergence. Since DRU essentially implements rank-based loss minimization, it aligns well with the concepts presented in Paper [2]. By integrating these theories, we plan to provide a theoretical sketch demonstrating the convexity of the evolving, rank-based loss objective function of the Discounted Rank Upweighting (DRU) procedure and provide a convergence argument in the revised version (as also requested by Reviewer 2).
>
> [1] Ren, Mengye, et al. "Learning to reweight examples for robust deep learning." International conference on machine learning. ICML, 2018.
>
>
> [2] Xiao, Rufeng, et al. "A Unified Framework for Rank-based Loss Minimization." Neural Information Processing Systems. NeurIPS, 2023.
>
> 2. Requirement of Additional Validation Dataset:
>  The use of a validation dataset is indeed common in machine learning to tune hyperparameters. We used the same setup (including the train/validation splits) for Amazon, Yelp, and IMDB datasets as suggested by the original authors of those datasets. That said, at your request, we performed some experiments testing the impact of validation dataset size on accuracy. It turns out that for IMDB/Yelp datasets we can shrink the size of the validation dataset without a significant decrease in accuracy. However, for the Amazon Dataset, the validation dataset should have similar size as the testing dataset; we saw some accuracy drop by reducing the validation dataset size. We will add these results to the revised version of the paper and discuss them.
>
>
>
> 3. Efficiency of the Proposed Method:
> Thanks for your question. Actually, it seems that there is some misunderstanding here. Our method (DRU) involves a single training (ERM) pass with multiple epochs, rather than multiple ERM passes. This is in contrast to the JTT approach, which requires training the model with ERM twice, each involving multiple epochs. The biggest cost in our method is attributed to the computation of rank and subsequent reweighting. Since this ranking is determined based on the groups, the time complexity for the rank and reweight process can be estimated as O(|g|log|g|), where |g| represents the number of groups. Hence, our approach is very computationally efficient and requires only a single pass over the training data. We will clarify this in the revised version of the paper.
>
>
>
> 4. Ablation Studies:
> We appreciate the suggestion for ablation studies on the number of validation data and computational efficiency. We will include these studies in our revised manuscript to provide a more comprehensive evaluation of our method.
>
> In conclusion, we are confident that these revisions will further strengthen our paper and appeal to the broader AI research community. Looking forward to your response!

---

### Review · Reviewer_Y93y · 2023-10-30

**Summary Of Contributions:**

The paper proposes to change the training and/or validation scheme in the context of out-of-domain generalization with group shifts (new groups in validation/test than in test). While the common practice is to minimize the worst-case group accuracy for validation or as a learning objective, and with the understanding that the mean tends to be unfavorable to small groups, the the authors propose to interpolate between the two using what they call a ``DCG'' (discounted cumulative gain) applied to group losses, which allows to interpolate between the max and the mean of group losses.
The authors propose an algorithm based on iterative reweithing of groups and perform experiments on standard benchmarks to validate their approach.

**Audience:**

No

**Claims And Evidence:**

No

**Requested Changes:**

see weaknesses above regarding presentation/execution

**Strengths And Weaknesses:**

strength:
- the idea makes sense, the worst-case nature of the selection or training may introduce some instabilities and a more generic criterion might be able to avoid that.
- the experiments show the validity of the approach, the baselines seem adequate

weaknesses:
- the paper is a bit minimalistic in terms of presentation and execution:
* What the authors call the DCG is in fact an ordered weighted averaging operator (see below). While I appreciate the honesty of the authors to clarify their source of inspiration, the analogy between learning to rank and the optimization of worst-case group accuracy is a bit shallow. The fact that their criterion is to minimize the DCG of losses, which reads as "minimizing a gain", illustrates the limitations of this analogy.
* I would have liked a convergence proof/argument for the algorithm. My understanding is that the objective function is convex and the iterative reweigithing boils down to taking a form of subgradient, so I would expect such arguments to be feasible in a reasonable amount of time
* the number of baselines is fairly large but the number of datasets fairly low (e.g., only the Amazon reviews dataset from the WILDS benchmark). I understand that the datasets are pretty large and making experiments is costly, so I would not rate it as a "requested" change but rather nice-to-have, since the value of the paper is moslty experimental.


== other comments
- application to fairness: what is an example where we have unseen groups that are not in training

comments:
* after Eq 2: ``This minimax approach to robust model selection is suboptimal since it ignores the predictive signal from other groups.'' -> I'm not sure what the authors mean by that, but I don't see how it can be true in general. For example, if g^* is the group with lowest loss in the min/max, then in general this loss is larger than the loss on g^* obtained by training on g^* only -- which means that other groups are taken into account.
* after eq 2: ``It also simplistically assumes that the worst-performing group on the validation dataset is
distributionally similar to the worst group on the OOD test set. S'' -> I'm not sure this assumption is made. I suspect the intuition of the min/max on the validation set would be that if you happen to generalize fairly well across all groups in the valid set, you should generalise across all groups in the distribution.
* after Eq3 : ``Since the inverse logarithm function'' -> to me the inverse logarithm is the exponential
* 4.1 I think it would be more exact to say that the authors use an ordered weighted averaging (OWA) operator (see e.g.,  https://en.wikipedia.org/wiki/Ordered_weighted_averaging_aggregation_operator and references there) rather than a DCG. Fundamentally, DCG@k is a function of a vector of score (denoted y) and a ranking (denoted sigma), and it gives
DCG(y, sigma) = sum_{i}w_i y_{sigma(i)}
where w_i are non-increasing, can be arbitrary but usually set to 1/log(i+1) if i<= k and w_i=0 as in the paper.
The authors here rather use an aggregation function of the loss per group
F(accuracies) = sum_{i} w_i loss_(i)
where loss_(i) is the i-th largest loss. written in DCG terms, F(accuracies) = max_{sigma} DCG(loss, sigma). Also notice that for DCG, higher is supposed to be better (G means "Gain") where here we want to maximize F. Rather, F is an OWA aggregation of losses. Notice that up to normalization, depending on weights w_i, OWA operators include the max, the mean and the min, and others in between. In the context of fairness, OWA operators are used in generalized Gini aggregation functions (see e.g., https://proceedings.mlr.press/v70/busa-fekete17a.html and https://dl.acm.org/doi/abs/10.1145/3477495.3532035)

---

> ### Author Response · Authors · 2023-11-23
> **Thank You. Clarifications and Revisions**
>
> Thank you for your detailed review and valuable insights. Sorry for the typographical errors, we will fix them in the revised manuscript. We  value your constructive criticisms, which we address as follows:
>
> 1. On the Use of 'DCG' Analogy & OWA operators:
> Thank you. Our choice of the term 'DCG' was indeed inspired by learning-to-rank literature, aiming to provide a familiar reference point for readers. In our revised manuscript, we will clarify that while our approach draws conceptual parallels to DCG, it more closely aligns with the principles of Ordered Weighted Averaging (OWA) operator in the context of group loss aggregation. This change will ensure a clearer and more accurate representation of our method.
>
> 2. Theory/Convergence:
> Please see our common response to all the reviewers above regarding theoretical justification.
>
> 3. Number of Datasets Used:
>   Our decision to utilize three real-world datasets (Amazon, Yelp, IMDB) in our experiments was driven by practical considerations. Given the large size and representativeness of these three text datasets, we consider them sufficient to evaluate the effectiveness of the proposed algorithm in the text domain (the main focus of our paper).  One another important  reason for selecting Amazon and Yelp from the WILDS benchmark (yes, Yelp is also part of WILDS benchmark, but is described in that paper’s Appendix) was our specific focus on the presence of numerous groups and particularly unseen ones in test data. This focus aligns with the core theme of our research. We emphasize that the DCG/OWA-type weighting approach is particularly effective in datasets with a rich diversity of groups. The proposed ranking-based methodology leverages the inter-relations (correlation structure) among the numerous groups in training. In cases where only a few groups are present (like many of the WILDS benchmark datasets), our approach might not be necessary.
>
> We are open to expanding our dataset range and would welcome any suggestions for publicly available text datasets that are sizable and have numerous groups.
>
> 4. Clarifications:
>
> >>>after Eq 2: ``This minimax approach….since it ignores the predictive signal from other groups....”
>
> Thanks. It seems that there is some misunderstanding. We did not mean that other groups did not contribute to the minimax approach. Of course, it leverages the correlational structure of the errors of the different groups, as the training phase takes examples in all groups and the identification of the worst group is also relative to the errors of other groups. Our point was that the objective that the minimax approach optimizes just relies on the worst performing group, and in the optimization process it often only upweights (or optimizes directly, as done by DRO or GroupDRO) the worst group (hence in that regard, it ignores the contribution of other groups). We will clarify this in the revised version of the manuscript.
>
> >>>after eq 2: ``It also simplistically assumes…dataset is distributionally similar to the worst group on the OOD test set.''....”
>
> We will also clarify this in the revised manuscript. Here again, we meant that the objective function that is optimized is based on the worst-group error which implicitly assumes that a similar error distribution would be present in the OOD test data. However, both validation and test datasets are OOD w.r.t training data, so this assumption could potentially be incorrect. Precisely like you said, the intuition is to find a model that generalizes fairly well *across all groups*, and a more reasonable way to deliver this intuition is to require the model to perform well on multiple low-ranked groups rather than the worst-performing group.
>
> 5. >>>application to fairness: what is an example where we have unseen groups that are not in training
>
> Here are a couple of real-world examples where unseen groups are part of testing data. One example is in sentiment analysis for social media, where a model trained on English posts from North America and Western Europe might later face posts from Southeast Asia or Africa. These regions use English differently, with unique slangs and expressions, presenting unseen language and dialect variations. Additionally, cultural context poses challenges. The model may encounter posts with cultural nuances or colloquial language from groups not included in the training data, like indigenous communities.
>
> Another example is in education (learning analytics). A model designed to personalize learning for students in a specific academic year may struggle to adapt to new students in subsequent years. These new students, with their distinct learning styles and backgrounds, form an unseen group. Ensuring the model's effectiveness for these students underscores the need for robustness against such group distribution shifts.
>
> In conclusion, we believe these revisions will address your concerns and strengthen our contribution to the literature. Looking forward to your response!

---

### Review · Reviewer_ufU7 · 2023-11-13

**Summary Of Contributions:**

This paper proposes two new methods for robust training and model selection in the presence of group distribution shift: it is assumed that the sampling distribution is a mixture of distributions over a set of groups where each of the children/within-group distributions are fixed but the distribution over the choice of group varies between the train, validation and test sets. The idea of both methods is to emulate the minimax strategy (minimizing the loss on the worst group) in a soft way.

For model selection/validation, the proposed model uses a performance measure defined just like the DCG in the ranking literature, but using the ranking of validation performance (from worst to best) as the corresponding ground truth ranking. At test time, the performance of the method is evaluated by comparing the ranking provided by this modified DCG and the ranking of the models to be evaluated by test performance.

For model training, the method consists in upweighting the samples which correspond to worse performing groups in the previous epoch via the formula on equation (6).

There are also variants of both models which use a modified version of DCG which stratifies the groups by quantiles, which is necessary when there are many groups, as is the case in the real data experiments.

In the real data experiments, the authors study the task of predicting the numerical rating of reviews provided by real users on three datasets (amazon, yelp, IMDB). Here, the "groups" are the "users", and the authors synthetize a modification of the datasets with OOD behavior by drawing separate users for validation and test set (users which were not sampled in the training set). The experimental results show mixed behavior generally tending towards the confirmation of the efficiency of the proposed method.

In synthetic data experiments, the authors create a simple toy dataset where each data point lies in $\mathbb{R}^2$ and is the sum of 2 terms: a "signal" term which is drawn from a fixed global distribution and determines the label, and a group specific "noise" term  which is drawn from a fixed distribution from each group, with that distribution being itself drawn independently for all groups through categorical sampling from a predetermined set of distributions. The results show significant superiority of the proposed method compared to the baselines.

**Audience:**

Yes

**Claims And Evidence:**

Yes

**Requested Changes:**

1. Could you explain both the precise procedure and the rationale behind the synthetic data experiments a little better?  What are the values of the $W$ with the predetermined noisy signals? What justifies the choice of $U=0.2$ and $U=0.8$? Did you experiment with varying values of $\sigma$?  How many groups are there in total? For instance, am I correct in assuming that there are 12 (i.e. 1+5+1+5) groups in Setting 1?

Honestly, instead of adding noise through the sigma and the sine function, it might make more sense to study the problem in a regression setting.



2. Please improve the clarify punctually as per my "minor comments" in the weaknesses section.

**Strengths And Weaknesses:**

Strengths:

1.This is an original idea with great potential practical application.
2. The real data experiments are quite interesting. This is an unusual way to construct an OOD dataset.

Weaknesses:

1. This is an absolutely purely applied paper with no theory whatsoever. Not even the problem setting is described in a mathematically formal way. There is no assumption about how the distributions across groups vary or how much the distribution over groups can vary between train and test sets.

2. The synthetic data experiments look a little bit contrived: the precise choice of all the parameters is so specific that it leaves a lot of room for cherry picking. To some extent, the same can be said of the real data experiments.





===========Minor comments ====
In page 2, the sentence “Further, the DCG metric is less prone to having ties in hyperparameter choices, leading to statistically identified models.” Seems a little confusing.
The quantile version of the DCG metric (DCG_q) is very unintuitive when presented to a reader in the form of equation (4) without any context from the experiments section: it turns out that this makes sense because the authors are interested in settings where there is a very large number of groups (the groups are the users in a recommendation system dataset). It might make sense to mention near equation (4) that it is mostly relevant to situations where there is a very large number of groups.

The paragraph which starts with the following sentence on page 7 is a little confusing: “Since we want to contrast with the hard minimax approach throughout this paper, we train models by only upweighting samples of the worst-performing group from the previous training epoch by a constant weight $\lambda$.”  Indeed, it seems to imply that that strategy is the only one used completely superseding the training strategy of gDRU and qDRU.

In the textual description of the synthetic data generation procedure, it is not clear that the threshold $a$ is fixed for the whole group, this can only be inferred from the algorithm in Appendix Section D. The description of the synthetic data experiments should be improved in general as well. Instead of writing something like “p: probability of each idiosyncratic signal”, the authors should consider introducing variables for the idiosyncratic signals. In addition, the definition or an introduction to the concept of "setting 1", "setting 2" etc. is needed in the main paper (I know that the description is in the appendix).


==============typos====================

Page 2, just below the bold part: ”First, we use DCG metric…” =>”First, we use the DCG metric…”

Period missing at the end of Equation (2), Equation (3), Equation (4) and Equation (5). Below equation (5), “Equation 5” should be “Equation (5)”. Similarly for “equation 7” a little bit below Equation (7).

In Equation (6), ``<= ” should be $\leq$
At the beginning of the experiments section: “intuitively, the performance… significantly downgrades than in ID validation…”=> “intuitively, the performance… significantly downgrades *compared to* the situation with ID validation…”

Bottom of page 7: “this variant will help us tease apart…” => “this variant will help us isolate….”

Page 14 “…so the t-stat should be the higher the better” => (for instance) “…, thus, a higher t-stat corresponds to a better model”

There is a coma instead of a forward slash in the second part of the experiments table just before Section 7 (second row, gDRU+G column).

---

> ### Author Response · Authors · 2023-11-23
> **Thank You. Clarifications and Revisions.**
>
> Thank you for your comprehensive review and the insights provided. We appreciate your recognition of the originality and potential practical application of our work, as well as your interest in our experimental details. Below is our response to your concerns:
>
> 1. Theory/Convergence:
> Please see our common response to all the reviewers above regarding theoretical justification.
>
> 2. Synthetic Data Experiments:
>  The parameters in the synthetic experiment were chosen to simulate different control conditions and are definitely not cherry-picked. The goal of the synthetic experiment is to demonstrate the validity of the proposed method rather than rigorously evaluate its performance (the latter is done with real-world datasets). In the revised version, we will present more detailed justifications for these design choices and include additional experiments with varied parameter settings.
>
>  It seems that there is a misunderstanding here. We don’t have only 12 groups in setting 1 as you mention. As stated in paragraph 3 of Section 7.2, we generate 1,000 training groups, 500 OOD validation groups, and 500 OOD test groups in each of our settings. As stated in paragraph 1 of Section 7.2, $W$ contains four idiosyncratic signals (gaussian distributions) with unit variance and mean being [0.25, 0.25], [0.25, -0.25], [-0.25, 0.25], and [-0.25, -0.25]. The mean vectors are simply chosen to ensure the four signals are symmetric and orthogonal to each other. Similarly, the value of $U$ is set as 0.8 or 0.2 simply to simulate scenarios where there are more or less distributional shifts in the test data. We did not try different values of $W$ and $U$ beyond those reported, and therefore we are not able to cherry-pick the results. We will report additional parameter settings in the revised version, but again the purpose of the synthetic experiment is really a demonstration of the validity of the algorithm rather than a rigorous evaluation.
>
> 4. Minor Comments and Typos:
>  We are grateful for your detailed observations on specific sections of the paper and the typos. We will carefully review and revise these sections for clarity and accuracy.
>
> Thanks once again for your insightful reviews. We believe that these revisions will substantially improve the manuscript and more effectively communicate the significance of our research. Looking forward to your response!

---

> > ### Comment · Reviewer_ufU7 · 2023-12-20
> >
> > Regarding the **synthetic data experiments**:
> >
> > I **thank** the authors for the clarification! I understand now that we can generate an arbitrary number of groups for each configuration/’setting’ and apologize for the oversight.
> > Thanks for adding more experiments with other parameters settings. I am reasonably convinced that there was no cherry picking. Still, I think studying a regression setting might be more natural.
> > Minor comment: in the algorithm (page 17), the authors might want to change “<=” to $\leq$.
> >
> >
> > Regarding **Section G**:
> >
> > I can't make enough sense of this section, it should be either rewritten more rigorously or deleted.
> >
> > Minor comments: after equation (9) on page 18, it would definitely make sense to mention the specific theorem in [1] which shows the result. In particular, the results from that paper are presumably specific to the ADMM optimization method, which should be specified in any Theorem here as well. I think the unnamed equation before eequation (9) is relatively improper notation. Indeed, by he authors’ own admission “ $n_iw_i$” is not fixed with $i$. This is true, but it means that a notation different from $n_i$ should be used. For instance, $n_{[i]}$ would make a lot of sense here. Further, although that is relatively minor, the results from [1] concern instance-level weights rather than group-level weights, which means that at least some notation should be fixed further below equation (9).
> >
> > I apologize if it is my fault for not understanding, but Section G seems far too informal. What do the authors mean by “$w_q$ is the weight for quantile q decided by the DCG function in our case”? (still on page 18). It seems they mean to talk about an alternative approach where the groups are exactly the quantiles. Using a proper formula would be better. Assuming my understanding is correct, the concept of $w_i$ is different in the first equation of the Section (where it means the weight associated with group $i$) and in the equations below it (where it means the weight associated with the corresponding quantile).
> > Also, what do the authors mean by “This is feasible if we keep the losses of each part of the group across boundaries the same in practice”: how can we “keep” the losses the same?
> > The whole argument seems contrived. If I understand correctly, the authors propose to calculate the weights first using a grouping based on quantiles, but then (because that wouldn’t be novel), recalculate weights for each group through an average of the quantile weights of all its elements. The final sentence which says that “we can prove the convergence of our method according to [1]” is not true. Indeed, they are arguing about the convergence of the modified method described in section G, not “their method”, i.e. the one used throughout the paper. Finally, the argument that the method described in Section G converges is just a handwave via the sentence “the loss of each quantile should be the same in expectation”. That statement is both vague and likely incorrect.  What is the expectation taken with respect to? In the example of Figure 2, it could be that the weightage for quantile 98 is var higher than that of quantile 99.  That would make the weightage of group $g_{[N-2]}$ far higher than that of $g_{N-1}$. It’s hard to see how any well-defined version of the statement “the loss of each quantile should be the same in expectation” could hold for quantile 99 in this sense.
> >
> > Note I still agree with reviewer Y93y that proving convergence might be possible without handwaving or modifying the algorithm too much, it might require slightly adapting some calculations from [1].
> >
> > ======================Reference=====================
> >
> > [1] A Unified Framework for Rank-based Loss Minimization. NeurIPS 2023

---

> ### Author Response · Authors · 2023-12-22
> **Thank You for your comments.**
>
> Thanks for your insightful comments.
>
> Regarding the convergence proof: We have revised and enhanced the rigor of Section G in the paper. The updated section G is in the revised PDF. That said, we want to clarify a few things regarding Section G. The fundamental “shape” of our proof is as follows:
>
> 1. **Uniform Group Sizes:** When groups are of equal size, then the objective function converges as per [1].
>
> 2. **Varying Group Sizes in Real Datasets:** However, in most real-world scenarios the group sizes are not uniform. Hence, we introduce a modified version of our DRU algorithm, termed “Modified qDRU” (see our previous response to “all the reviewers”) to apply the ideas from the proof of [1] to our case. In summary, we can only prove convergence of this modified qDRU approach and not our original DRU algorithm. And, to use the ideas from [1] to prove the convergence of modified qDRU, we require uniform quantile sizes and also need a sorted list of the average quantile losses. This necessitates partitioning the instances within a group to maintain equal average loss across partitions (recall that a group’s instances can be split across quantile boundaries), a task that is NP-hard (could be reduced to the Partition Problem).
>
> 3. **Practical (Data-Driven) Implementation Strategy:** We side-step this NP-hard problem of splitting instances with equal average losses by a weighted reweighting of the instances. This approach is feasible in real-world scenarios as it only requires knowledge of the number of instances in each partition/split which is readily available given that the sizes of both the groups and quantiles are known. Further details and implications of this approach are elaborated in the updated Section G. Please read the updated Section G and let us know if something is still unclear.
>
>
>
> >I apologize if it is my fault for not understanding, but Section G seems far too informal. What do the authors mean by “is the weight for quantile q decided by the DCG function in our case”? (still on page 18). It seems they mean to talk about an alternative approach where the groups are exactly the quantiles. Using a proper formula would be better. Assuming my understanding is correct, the concept of  is different in the first equation of the Section (where it means the weight associated with group) and in the equations below it (where it means the weight associated with the corresponding quantile).
>
> Yes, the weighting is still decided as in Equation 6 of the main paper. We apologize for the previous confusion caused by the description in Section G. In our approach, each quantile can contain multiple groups, and similarly, a single group can extend across multiple quantiles. However, it is crucial that every instance within a given quantile is weighted in accordance with Equation 6 from the main paper. The designation of quantile numbers, such as 98 or 99, represents a normalized ranking, which is calculated by dividing the total number of groups.
>
>
> >Also, what do the authors mean by “This is feasible if we keep the losses of each part of the group across boundaries the same in practice”: how can we “keep” the losses the same? The whole argument seems contrived. If I understand correctly, the authors propose to calculate the weights first using a grouping based on quantiles, but then (because that wouldn’t be novel), recalculate weights for each group through an average of the quantile weights of all its elements. The final sentence which says that “we can prove the convergence of our method according to [1]” is not true. Indeed, they are arguing about the convergence of the modified method described in section G, not “their method”, i.e. the one used throughout the paper.
>
>
> Apologies again for the confusion caused by the initial description in Section G. We acknowledge the challenge in proving the convergence of our original method as outlined in the main body of the paper. Instead, we focus on showing the convergence of a slightly modified method, "modified qDRU." Our claim centers around the premise that if we can partition a group across quantiles such that average losses of instances in each partition is the same, then, we can use the proof of [1]. While achieving such partitioning is NP-hard theoretically, in real-world settings, this can be effectively addressed through weighted reweighting which side-steps solving the NP-hard partition problem. In the updated Section G, we have illustrated how these recalculated weights make the practical implementation equivalent to the theoretical model.

---

### Author Response · Authors · 2023-11-23
**Theoretical justifications/Convergence arguments.**

Dear Reviewers,

Since you raised some concerns regarding theoretical justifications of the proposed approach/convergence guarantees, etc, here's our common response to all of you.

 We appreciate the suggestion to include a formal theoretical analysis, as it would indeed deepen the understanding of our proposed algorithm. Our work primarily aims to introduce and empirically validate a novel algorithm tailored for real-world applications, particularly in scenarios where group distributional shifts are present. This focus aligns with similar studies in the field, notably the Just-train-twice (JTT) paper (ICML 2021), which has been a significant inspiration for our approach. While we acknowledge the value that a theoretical analysis would add, it's important to note that the absence of such analysis does not diminish the practical applicability of our algorithm. This perspective is exemplified by the JTT paper, which has demonstrated substantial practical value despite the absence of formal theoretical results. Nevertheless, we agree that further theoretical exploration could be a beneficial direction for future work and will consider this in our ongoing research.

In DRU, reweighting is performed every epoch based on the group loss observed during that epoch. This means the objective function of the algorithm changes over epochs because the group weights are dynamically adjusted based on each group’s loss, which increases the complexity of analyzing the optimization procedure precisely. That said, there is theoretical work that can be related to the convergence of our proposed algorithm, in particular the methodologies outlined in recent papers [1,2]. Specifically, Paper [1] establishes the convergence of a reweighting algorithm that adjusts individual sample weights based on gradients. Paper [2] introduces a comprehensive framework for rank-based loss minimization and discusses strategies to achieve convergence. Since DRU essentially implements rank-based loss minimization, it aligns well with the concepts presented in Paper [2]. By integrating these theories, we plan to provide a theoretical sketch demonstrating the convexity of the evolving, rank-based loss objective function of the Discounted Rank Upweighting (DRU) procedure and provide a convergence argument in the revised version (as also requested by Reviewer 2).

[1] Ren, Mengye, et al. "Learning to reweight examples for robust deep learning." International conference on machine learning. ICML, 2018.

[2] Xiao, Rufeng, et al. "A Unified Framework for Rank-based Loss Minimization." Neural Information Processing Systems. NeurIPS, 2023.

---

### Author Response · Authors · 2023-12-08
**Thank You. We are submitting the updated PDF.**

Dear Reviewers and Action Editor,

Thanks for providing the extension to submit the revised PDF. The reviews came at a really busy time of the year with many holidays and end-of-semester tasks, so it took us longer than usual to submit our response and the updated PDF. More importantly, we also wanted to properly address all your concerns; hence the delay. Once again, thanks for all your comments and suggestions. They have definitely helped us improve the paper significantly. All the changes in the revised version of the paper are highlighted in blue text. Here are the key changes we made:

1). We fixed all the typos.

2). We added results with varying levels of validation data as requested by Reviewer re2Y. They are in the Appendix.

3). We discuss the computational complexity of our approach w.r.t ERM, JTT, GroupDRO, as requested by Reviewer re2Y

4). We clarified the details of the synthetic data experiments as requested by Reviewer ufU7.

5). We performed extra synthetic data experiments as requested by Reviewer ufU7. We experimented with values of sigma =1 and 2. The results are in the Appendix. (The resulting trends are the same– DRU approaches significantly outperform the baselines)

6). We added our approach’s connection to OWA as a footnote in Section 4.1. Thanks again to Reviewer Y93y

7). Finally, we added some convergence theory (requested by all the reviewers) for our approach in Section 4.2.1 and in the Appendix. It turns out that there is a recent paper by Xiao et al. 2023 which considers a ranked-loss minimization setup as ours. We can directly apply their convergence arguments to our case when the group sizes are the same. When the group sizes are different, we have to alter our qDRU algorithm a little to fit it into the framework of Xiao et al. 2023 and show convergence. Empirically,  the modified qDRU algorithm performs better than ERM, but slightly worse than qDRU in terms of worst group accuracy. We haven’t put those full results in the paper, but we can if the review team thinks so.

| Model         | Yelp (average/10th/worst) | IMDB (average/10th/worst) |
|---------------|----------------------------|----------------------------|
| ERM           | 63.0/52.0/18.0             | 63.2/42.9/15.0             |
| Modified qDRU | 62.5/52.0/22.0             | 61.5/43.8/16.1             |
| Qdru          | 62.6/53.0/23.0             | 64.1/45.8/20.0             |

---

### Author Response · Authors · 2023-12-22
**Modified Section G (Appendix) is in the updated PDF**

Unfortunately, the text was too long to fit as a comment here, so the updated convergence proof as per the request of  Reviewer ufU7 has been provided in the revised PDF. It's section G in the revised PDF.

---

### Decision · Action_Editor_i5c2 · 2024-01-22

**Recommendation:** Reject

**Comment:**

To begin, the reviews for this paper are by no means enthusiastic. Even after a solid rebuttal and discussion period including revisions of the paper, two of three reviewers were left highly skeptical of whether or not this paper meets the acceptance threshold of TMLR.

One reviewer says: *"All in all, the quality of the execution of the paper was poor and the modifications to the paper nowhere near what is required to meet high quality standards."*

Another reviewer states: *"Section G (which was added in response to requests from another reviewer) is still quite confusing to me and I cannot vouch for its correctness, despite multiple rounds of discussion with the authors."*

Having looked over the paper myself, I find myself in agreement with these reviewers. I found the basic formulation of both the underlying problem and the proposed methods quite unclear, and rife with ambiguity, making it very difficult to assess the validity of the more technical points underlying the authors' proposal. I am not a reviewer, and I may be missing some points, but for the future reference of the authors, I will highlight a handful of points that I tripped up on below.

- The ultimate goal is buried in the text. Paragraph 1 of section 3 leads the authors to think we are interested in some general loss, but then in paragraph 2 of the same section they say "test set accuracy" is the quantity of interest. This is even further complicated by the statement that *"this paper assumes no group overlap between OOD test and training/validation sets"*. I have no idea why the authors make this statement, since it always leaves open the possibility that groups at test time have distributions that are totally unrelated to those sampled at training time, rendering the problem meaningless. If the distance between distributions played some role in the subsequent exposition, it would be more understandable, but such a narrative was not apparent to me.

- In section 4, up to the end of sub-section 4.1, the focus is on *model selection*, and yet for some reason the authors formulate their proposed metric in terms of $l\_{g}$, which is the expected (group $g$) *surrogate* loss. As a training objective, I of course understand the use of a surrogate for optimization reasons, but the authors state in several places throughout the paper that their interest is in *accuracy* at test time. If we just compute and compare metrics over model candidates (assumed to be already trained), then why not use accuracy? If found this quite puzzling (especially in conjunction with the exposition of 4.2).

- In sub-section 4.1, the authors say *"Next, sort all the $m_{val}$ groups according to their loss..."* but exactly what is being sorted is totally unclear, and left abstract throughout the entire paper (to the best of my reading). If we are assigning order to *groups*, then we need one representative value for each group. Each group can have more than one point, and thus more than one loss value; this makes the *"according to their loss..."* bit totally ambiguous. Should it be the average loss? Presumably so, but this is not stated.

- As pointed out by one of the reviewers, the use of the word "quantile" is *highly* misleading (starting in last paragraph of 4.1, but continuing elsewhere). The authors talk about *"groups at quantile 0 (worst-group), quantile 1, up to quantile $k$"*. What is being ordered here? Picking up from my previous point, if we are sorting average within-group losses, then the resulting quantiles are going to either be sorted (average within-group loss) values, or values between these in the event of ties; using "quantile" to refer to order *indices* $0, 1, 2, \ldots$ is misleading.

- Regarding the convergence of DRU, I don't want to get into too many details because one reviewer already went back and forth with the authors several times, but I also found this exposition quite confusing. I know that these details were added in response to one reviewer, but it really feels like a tacked-on section. Xiao et al. (2023) assumes a linear model to leverage convexity; not even this point is mentioned in the main text.

Overall, I think there are some solid ideas underlying the submission here, but I have to agree with the reviewers that the presentation of those ideas is below the standard expected for TMLR.

**Audience:**

This is a very general problem, capturing a wide variety of machine learning problem settings, including fairness-aware classification and classification under distribution shift. The problem of interest is definitely one in which many other machine learning researchers are interested, and thus in principle, the audience for this paper should be quite wide. Clarity issues, however, render the effective audience quite narrow, I believe. Please see additional comments for more on this point.

**Claims And Evidence:**

The basic problem setting of this work (as I understand it from the first two paragraphs of section 3) is a classification task, in which the data may be sampled from distinct "groups," corresponding to distinct probability distributions, and the ultimate goal is to ensure that the accuracy on the *worst* group at test time is minimized.

The starting point for the authors' contributions is the claim that the usual minimax approach (which ignores all but the worst group) is sub-optimal. Their reasoning is centered on the following point: the worst group at training time may not be the worst group at test time, and by ignoring all but the worst group can waste a lot of data that could be beneficial to improve worst-group accuracy at test time.

While the notion of "sub-optimal" is somewhat obscure, the underlying intuition is clear, and the authors suggest utilizing a "softened" version of the minimax objective, which takes a weighted sum of the relevant metrics computed for the $k > 1$ worst groups. This is their "DCG" metric, positioned in this paper as a tool for *model selection* (i.e., selection of trained models).

To complement the DCG metric for model selection, the authors also describe a strategy for *training* (their sub-section 4.2; method called DRU). Essentially, given an iterative training algorithm, at each step or epoch $t$, the authors update the weights used in a weighted sum (over groups) of average surrogate losses (see their equation 7). The update places more weight on worse-performing groups in terms of *accuracy* (not surrogate losses).

At a high level, the motivations and proposed methods of the authors is clear, and experimental evaluation illustrates some settings in which the proposed approach does indeed offer a better balance between worst-case group accuracy and average accuracy.

On the other hand, the concrete formulation of the rough ideas described above is in my opinion quite sub-standard, and I think parsing this paper will be a challenge for most potential readers, even those which are very familiar with the related literature. Please see additional comments for more on this point.

**Resubmission Of Major Revision:**

The authors may consider submitting a major revision at a later time.